# The Sexual Dimorphism in Cerebrospinal Fluid Protein Content Does Not Affect Intrathecal IgG Synthesis in Multiple Sclerosis

**DOI:** 10.3390/jpm12060977

**Published:** 2022-06-16

**Authors:** Massimiliano Castellazzi, Caterina Ferri, Ginevra Tecilla, André Huss, Paola Crociani, Gaetano Desina, Gianvito Barbella, Alice Piola, Samantha Permunian, Makbule Senel, Maurizio Leone, Hayrettin Tumani, Maura Pugliatti

**Affiliations:** 1Department of Neurosciences and Rehabilitation, University of Ferrara, 44121 Ferrara, Italy; caterina.ferri@unife.it (C.F.); ginevra.tecilla@unife.it (G.T.); gianvito.barbella@unife.it (G.B.); alice.piola@edu.unife.it (A.P.); samantha.permunian@edu.unife.it (S.P.); maura.pugliatti@unife.it (M.P.); 2University Center for Studies on Gender Medicine, University of Ferrara, 44121 Ferrara, Italy; 3Department of Neurology, University Hospital Ulm, 89081 Ulm, Germany; andre.huss@uni-ulm.de (A.H.); makbule.senel@uni-ulm.de (M.S.); hayrettin.tumani@uni-ulm.de (H.T.); 4Neurology Unit, Fondazione IRCCS “Casa Sollievo della Sofferenza”, 71013 San Giovanni Rotondo, Italy; pcrociani@alice.it (P.C.); ninodesina@tiscali.it (G.D.); m.leone@operapadrepio.it (M.L.); 5Interdepartmental Research Center for the Study of Multiple Sclerosis and Inflammatory and Degenerative Diseases of the Nervous System, University of Ferrara, 44121 Ferrara, Italy

**Keywords:** multiple sclerosis, cerebrospinal fluid, intrathecal IgG synthesis, blood-cerebrospinal fluid-barrier, sex, age

## Abstract

(1) Background: Multiple sclerosis (MS) is a chronic inflammatory demyelinating disease of the central nervous system (CNS) that mainly affects young adults and females more than males. The detection of intrathecal IgG synthesis (IIS) on cerebrospinal fluid (CSF) analysis supports the diagnosis of MS. A sexual dimorphism has recently been described in CSF protein content. (2) Methods: Clinical and laboratory data from 340 MS patients (F = 231, M = 99) and 89 people with clinically isolated syndrome (CIS) (F = 57, M = 32) were retrospectively analyzed to assess the presence of variables affected by sex and age. (3) Results: In MS, the albumin quotient (QAlb), reflecting the blood–CSF barrier (BCSFB) function, was higher in males (5.6 vs. 4.34) and correlated to age with a constant difference between sexes (F = 41.71). In CIS patients, QAlb increased with age only in males (*r* = 0.3567). Age was positively correlated to disease duration and severity in MS (*r* = 0.3502, *r* = 0.2986, respectively). No differences emerged for quantitative and qualitative IIS determinations. (4) Discussion: Although the main difference between males and females concerns the function of BCSFB assessed by QAlb, this sexual dimorphism does not affect the determination of the IIS evaluated both by quantitative and qualitative methods.

## 1. Introduction

Multiple sclerosis (MS) is a chronic inflammatory demyelinating and neurodegenerative disease of the central nervous system (CNS) [1]. It mainly affects young adults, with onset between the ages of 20 and 30, and is the leading non-traumatic cause of neurological disability in young people [2]. MS is predominant in the female sex with a female-to-male ratio ranging between 2:1 to 3:1 depending on the geographic region [3]. This ratio becomes greater with age due to the increased disease incidence in women but not in men [4]. 

Almost 85% of people with MS have a relapsing-remitting course characterized by clinical relapses alternating with phases of well-being [5]. Most of the patients will develop a secondary progressive form of disease across years with a gradual accumulation of disabilities [6]. Besides the MS distribution, sex has also been related to prognostic aspects. Women are more prone to inflammatory relapses compared to men, who in return show more frequent signs of disease progression or brain atrophy [3,7].

The first clinical manifestation of MS is known as CIS (clinically isolated syndrome), which by definition implies a monophasic condition [8]. MS diagnostic criteria are based on the presence of demyelinating lesions of the CNS and clinical symptoms with spatial and temporal dissemination. According to the latest McDonald criteria revision in 2017, the finding of immunoglobulin G (IgG) oligoclonal bands (OCB) in the cerebrospinal fluid (CSF) has been included as an additional criterion allowing the diagnosis even in the absence of clinical or neuroradiological temporal dissemination [6]. CSF analysis, therefore, represents an important evaluation during the diagnostic phase of the disease [9].

CSF is a watery fluid that permeates the entire CNS. It is mainly produced in the choroid plexuses of the cerebral ventricles and is largely similar to a plasma ultrafiltrate in that most CSF proteins are plasma-derived [10,11]. For decades, CSF analysis has had an important role in the management of the neurological patient and, in particular, in a patient with suspected MS [12]. Among the diagnostic findings that can be obtained from CSF analysis, there is quantitative evaluation of the blood–CSF barrier (BCSFB) function using the albumin quotient (QAlb) and quantitative and qualitative determination of intrathecal antibody production, both through calculation of mathematical indexes and the demonstration of IgG OCB, respectively [13].

In recent years, the existence of sexual dimorphism at the level of CSF protein content has become increasingly evident, and in particular, these sex-related differences seem to concern all the concentrations of plasma-derived proteins [14,15,16,17]. It has emerged that QAlb is higher in males than females regardless of age as a consequence of a reduced CSF-flow rate (e.g., an increased transfer of protein from blood to CSF, reduced CSF elimination from CSF space into lymphatic organs or venous compartment) in the male sex. These data were confirmed in neurological patients [16], psychiatric subjects [18], as well as in healthy individuals [19]. This increase in QAlb levels means that male patients are more likely to receive a laboratory report suggesting altered CSF protein content and functional impairment of BCSFB with respect to females [14,16,19]. Moreover, the difference in CSF protein content found in the two sexes did not only concern albumin, the most concentrated protein in CSF, but also IgG, the second most abundant protein, as demonstrated in a population of neurological patients negative for intrathecal IgG synthesis (IIS) [16,17]. Furthermore, it has been shown that in addition to sex, aging can also affect CSF protein content and the BCSFB function [12].

In light of the discrepancies between males and females in the onset and course of multiple sclerosis and of the effects of sex and age on CSF protein content and BCSFB permeability, this study aims to investigate whether sex and age have an impact on CSF analysis used to support the diagnosis of MS.

## 2. Materials and Methods

### 2.1. Study Design

Anonymized clinical and laboratory data were collected retrospectively from patients admitted to the three centers participating in the study: (i) Ferrara, northern Italy; (ii) San Giovanni Rotondo, southern Italy; and (iii) Ulm, Germany. The study was approved by the Medical Ethics in Research Committee of the coordinating center of Ferrara, “Comitato Etico di Area Vasta Emilia Centro della Regione Emilia-Romagna” (Prot. N. 770/2018/Oss/AOUFe, 12 December 2018). Sampling, analysis, and data collection were performed with the approval of the respective local ethics committees in all participating centers, and informed consent was obtained from each patient upon admission.

All pre-analytical and analytical procedures were performed according to good clinical practice and following international guidelines [12,20,21]. The data collected concerning clinical and instrumental laboratory tests were used for diagnostic purposes without determining additional procedures for the patients and the health systems. 

The “time of the study”, for all patients, refers to the moment they underwent lumbar puncture for diagnostic purposes.

### 2.2. Patients

Three different cohorts of patients were included in this study.

Cohort 1 included patients hospitalized from 2005 to 2017 at the “S. Anna” University Hospital (Azienda Ospedaliero-Universitaria S. Anna), in Ferrara, northern Italy. Data from 266 patients with a definitive diagnosis of MS were included in the study. Patients with CIS were not included in this cohort.

Cohort 2 included patients hospitalized from 2013 to 2020 at the Hospital “Casa sollievo della sofferenza” in San Giovanni Rotondo (FG), southern Italy. Seventy-six MS patients were identified, and data were collected for the study. Moreover, in this case, CIS patients were excluded.

Cohort 3 included patients hospitalized from 2004 to 2011 at the Department of Neurology, University Hospital Ulm, Ulm, Germany. Data from 89 CIS patients were collected and included in the study.

None of the patients included in the study had received treatment with immunosuppressants or immunomodulatory drugs, including steroids, at the time of sampling.

### 2.3. Clinical Evaluations

For MS patients, relapse was defined as the onset of new or recurrent signs or symptoms or the worsening of already present neurological abnormalities persisting for at least 24 h in the absence of fever and preceded by at least 1 month of stable or improved neurological state [1]. The presence of relapse at the time of lumbar puncture was considered a clinical disease activity. Disease severity was scored through the use of Kurtzke’s expanded disability status scale (EDSS) [22]. For all MS and CIS patients, disease duration was defined as the time elapsed between clinical onset and study time.

### 2.4. Cerebrospinal Fluid Analysis

Paired CSF and serum samples were obtained during hospitalization as a part of the diagnostic work-up. The quantitative analysis consisted of the evaluation of the BCSFB permeability with the use of QAlb and in the determination of an IIS through mathematical indexes. 

Albumin quotient was calculated according to the formula: QAlb = [albumin]CSF/[albumin]serum × 1000

Normal QAlb values were considered as <6.5 for patients aged 15–40 years, <8.0 for patients aged 41–60 years, and <9.0 for patients over 60 years [12,23,24]. Accordingly, QAlb was considered abnormal for values greater than or equal to the reported thresholds.

Quantification of the IIS was done with both linear and hyperbolic mathematical formulas as follows:linear IgG index = QIgG/QAlb,(1)
where QIgG = [IgG]_CSF_/[IgG]_serum_ × 1000;
the hyperbolic Reiber’s formula, IgG_Loc_ = (QIgG − QLim) × [IgG]_serum_,(2)
where QLim $=0.93\times \sqrt{QAlb^{2}+6\times 10^{-6}-1.7\times 10^{-3}}$.

To calculate the IgG intrathecal fraction (IF)
IFIgG=(1−QLim/QIgG)×100 (%).

The Normal limits were considered 0.7 and 0 for IgG index and IF, respectively.

Qualitative analysis consisted of the research of IgG OCBs in CSF with the use of the “gold standard” of isoelectric focusing on an agarose gel followed by IgG-specific immunofixation. The following CSF patterns were considered: pattern 1 = absence of IgG OCBs; pattern 2 = 2 or more CSF-restricted IgG OCBs; pattern 3 = 2 or more CSF-restricted IgG OCBs with additional identical IgG OCBs in CSF and serum; pattern 4 = 2 or more identical IgG OCBs in CSF and serum; and pattern 5 = some identical IgG OCBs in CSF and serum in a restricted pH range. Only patterns 2 and 3 were considered suggestive of an intrathecal IgG synthesis [12]. 

### 2.5. Statistical Analysis

After checking for normality with the Kolmogorov–Smirnov test, the majority of continuous variables presented a non-normal distribution (except for age in males with MS and CIS, age, and QAlb in females with CIS). A non-parametric approach was adopted: data were reported as the median and interquartile range (IQR), and Mann–Whitney test was used for all comparisons. Categorical variables were reported as counts (percentages), and the Fisher’s exact test was used to test significance. Correlations were investigated with the Spearman test. In regression analysis, F-test was used to compare the fits of linear models. The agreement with quantitative and qualitative methods for the determination of an IIS was calculated through the Cohen’s kappa coefficient with a free available tool [25]: https://www.graphpad.com/quickcalcs/kappa1/ (accessed on 24 May 2022).

Kappa values were interpreted as follows: 0.00–0.20, slight agreement; 0.21–0.40, fair agreement; 0.41–0.60, moderate agreement; 0.61–0.80, substantial agreement; and 0.81–1.00, almost perfect agreement. Two-tailed *p*-values < 0.05 were considered statistically significant. Prism 9 for MacOS (GraphPad Software, La Jolla, CA, USA) was used for the statistical analysis.

## 3. Results

### 3.1. Patient Characteristics

The demographic and clinical characteristics of the study population are reported in Table 1. 

Cohort 1 and cohort 2 included patients with MS, while cohort 3 included patients with CIS. At the study time, there were no differences between males and females within the three cohorts of patients for age, clinical activity, disease duration, and severity (all *p* > 0.05). The same variables showed no sex-related differences between MS patients in cohort 1 and 2 (all *p* > 0.05). Owing to this lack of differences, cohorts 1 and 2 were merged in all subsequent analyses.

### 3.2. Cerebrospinal Fluid (CSF) Characteristics

CSF characteristics of the study population are reported in Table 2. 

QAlb value was higher in males than in females with MS (5.6 vs. 4.34), and the percentage of people with an altered QAlb, suggestive of BCSFB impairment, was greater in male MS patients compared to females (33.0 vs. 10.4%). There were no differences between male and female patients for the presence of an IIS using either mathematical formulas or the research of CSF-restricted IgG OCB in MS and CIS subgroups. The comparisons between MS and CIS subjects, grouped by sex, showed higher QAlb values in females with CIS than in MS (5.2 vs. 4.34). Other comparisons between MS and CIS groups did not reach a statistical significance for the quantitative and qualitative determination of an IIS.

### 3.3. Effect of the Blood-Cerebrospinal Fluid-Barrier (BCSFB) Permeability on Disease Severity and Duration

To investigate a possible role of an altered BCSFB permeability, MS patients were grouped by the presence or absence of an altered QAlb, and differences in EDSS and disease duration were checked in patients analyzed as a whole and grouped by sex. As reported in Figure 1, no differences emerged in the disease severity and duration at the study time by grouping MS patients by presence or absence of BCSFB dysfunction (all *p* > 0.05).

### 3.4. Effect of Age on Disease Severity and Duration

In MS patients, age, expressed in years, was correlated to disease duration and disease severity, scored by EDSS (Table 3). 

Age positively correlated to disease duration in MS patients analyzed as a whole (Spearman; *r* = 0.3502 and *p* < 0.0001) in male (Spearman; *r* = 0.4219 and *p* < 0.0001) and in female subgroups (Spearman; *r* = 0.3137 and *p* < 0.0001). Age was also positively correlated to EDSS in the whole MS group (Spearman; *r* = 0.2986 and *p* < 0.0001) in male (Spearman; *r* = 0.1997 and *p* = 0.0374) and female subgroups (Spearman; *r* = 0.3395 and *p* < 0.0001). EDSS and disease duration were positively correlated to each other in the overall population and in female subgroup (Spearman: *r* = 0.2837 and *p* < 0.0001, *r* = 0.3798 and *p* < 0.0001, respectively), while they did not correlate in male subgroup.

### 3.5. Effect of Age on Quantitative Cerebrospinal Fluid Indexes

In MS and CIS patients, age, expressed in years, was correlated to QAlb as an expression of the BCSFB permeability and to two mathematical indexes for the quantitative determination of an IIS: the linear IgG index (cut-off value = 0.7) and Reiber’s hyperbolic formula were used to calculate the intrathecal fraction (IF) (cut-off value = 0%). 

As reported in Table 4, age positively correlated to QAlb in MS patients analyzed as a whole (Spearman; *r* = 0.1807 and *p* = 0.0008) and in male (Spearman; *r* = 0.2711 and *p* = 0.0043) and female subgroups (Spearman; *r* = 0.1540 and *p* = 0.0192). In a simple linear regression model, the regression line was constantly higher in male than in female MS patients (differences between slopes: F = 1.849, *p* = 0.1748; difference between Y-intercepts: F = 41.71, *p* < 0.0001). Age was also positively correlated to QAlb in CIS patients analyzed as a whole (Spearman; *r* = 0.2368 and *p* = 0.0255) and in the male subgroup (Spearman; *r* = 0.4561 and *p* = 0.0087). In CIS patients, simple linear regression analysis was not performed due to the lack of correlation between age and QAlb in the female subgroup. A weak statistical significance was found for a positive correlation between age and the linear IgG index in male CIS patients (Spearman; *r* = 0.3567 and *p* = 0.0451). All other correlations between age and quantitative indexes of IIS did not yield statistically significant results in both MS and CIS patients (all *p* > 0.05). 

### 3.6. Impact of Sex and Age on IgG Oligoclonal Bands Determination

The “gold standard” of the CSF-restricted IgG OCB was used to assess the presence of an IIS in MS and CIS patients. As reported in Figure 2, by grouping patients by sex and comparing positivity for OCB in both MS and CIS patients analyzed as a whole or divided by age, no differences were found between males and females for OCB positivity (all *p* > 0.05). While grouping patients by the presence or absence of CSF-restricted OCB, there were no differences in age in MS and CIS patients analyzed as a whole and grouped by sex (data not shown).

### 3.7. Impact of Sex on the Agreement between Quantitative and Qualitative Methods for the Determination of an Intrathecal IgG Synthesis

The performance of the two mathematical indexes for the quantitative determination of an IIS, the linear IgG index (cut-off value = 0.7) and Reiber’s hyperbolic formula (IF cut-off value = 0%), were determined in MS and CIS patients (Table 5). 

For both formulas, the “gold standard” of the CSF-restricted IgG OCB was used to calculate: (i) concordance through the Cohen’s kappa, (ii) sensitivity, (iii) specificity, and (iv) positive and (v) negative predictive value (PPV and NPV, respectively). 

In MS patients, the IgG index reached a moderate agreement with the gold standard only in females (kappa = 0.407), but in general, both indexes showed a slight agreement or a fair agreement with the gold standard (kappa range: 0.307–0.371 for IgG index; 0.234–0.385 for the Reiber’s formula). In CIS patients, concerning the gold standard, both indexes showed a slight agreement in all subgroups (kappa range: 0.087–0.180 for the IgG index; 0.067–0.168 for the Reiber’s formula). In MS patients, the average sensitivity of the IgG index and the Reiber’s formula was 0.6699 and 0.6241, respectively, while in CIS subjects, they were 0.4749 and 0.4338, respectively. In both MS and CIS patients, the specificity of the two indexes was always equal to or greater than 0.9 (range: 0.900–1.000 and 0.925–1.000 for the IgG index and the Reiber’s formula, respectively), and the PPV was never less than 0.97 (range: 0.9714–1.000 and 0.9774–1.000 for the IgG index and the Reiber’s formula, respectively), while the NPV showed lower value, especially in the CIS population (range: 0.2941–0.3956 and 0.2500–0.3776 for the IgG index and Reiber’s formula, respectively in MS; 0.1053–0.1935 and 0.09524–0.1875 for the IgG index and the Reiber’s formula, respectively in CIS).

## 4. Discussion

In this study, we applied, for the first time, a sex-specific approach to a retrospective study of the laboratory results of CSF analysis performed to support the diagnosis of MS. If, on the one hand, our results substantially confirm that the main difference between males and females concerns the BCSFB function assessed by QAlb, on the other hand, they demonstrate that this sexual dimorphism does not affect the determination of the intrathecal IgG synthesis evaluated both by quantitative and qualitative methods. Moreover, our results show that age has a different effect on the increase in BCSFB function in the two sexes, with a greater impact in males than females in both MS and CIS patients.

Gender medicine is considered the study of how diseases differ between men and women in terms of prevention, clinical symptoms, therapeutic approach, prognosis, and psychological and social impact and is a relatively recent cultural approach that is not yet fully applied. This way of carrying out biomedical research has made it possible to highlight differences that were previously existent but which had not yet been taken into consideration. In the case of CSF analysis for the classification of the neurological patient, a sex-specific approach to laboratory data has, in recent years, shown a greater permeability of BCSFB in males compared to females with a consequent increase in protein transferred from the blood to CSF [14,15,16,17,18,19].

Irrespective of the underlying pathophysiology, which remains to be clarified, this sexual dimorphism impacts daily clinical practice. In fact, male patients are likely to receive a laboratory report suggesting altered BCSFB function three times more often than females regardless of the disorder they suffer from [16].

CSF analysis is a widely used laboratory tool to support the diagnosis of neurological disorders and plays a particularly important role in supporting the diagnosis of MS [5]. Quantitative CSF analysis, through the measurement of albumin and IgG in paired serum and CSF samples and the subsequent calculation of mathematical indexes, can provide information on the functionality of BCSFB and on the presence of inflammation within the CNS [20,21,26]. An increase in total CSF protein as well as an increase in QAlb are not typical features of MS patients, and elevated QAlb values greater than 20 are less likely to be compatible with the diagnosis of MS [12]. On the other hand, an increase in QAlb has been associated with greater brain atrophy and is considered a predictor of greater disability in MS [27]. In our study, by grouping MS patients based on the functional status of the BCSFB, we did not highlight any difference in the duration of the disease and its degree of disability as expressed by EDSS. This discrepancy between our data and the study published by Uher and colleagues is essential because we did not collect and then analyze magnetic resonance imaging (MRI) data in our patients [27]. Furthermore, since ours was an observational study referring to the time of diagnosis, no follow-up data were collected and analyzed. These data could have allowed us to carry out evaluations on the long-term impact of the altered BCSFB permeability. It is known that male patients tend to develop a more aggressive form of the disease [3,7]. In light of our data and those previously published [27], this increased disease severity could partly account for the fact that elevated QAlb values are more frequent in males and that these values are both associated with greater CNS tissue damage and a worse prognosis [27]. We do not know what the biological cause of this sexual dimorphism is, and the present study is not intended to be a mechanistic approach. However, the evidence and hypotheses accumulated to date seem to suggest two main possibilities: (1) on the one hand, a protective role of estrogen hormones on BCSFB function [28,29,30] and (2) on the other hand, much more simply, the fact that these discrepancies are due to the different height of the male population compared to the female one with a consequent increase in the rostro-caudal protein concentration gradient within CSF [18,31,32]. However, this remains a key point to understand because, in both cases, we may either be overestimating the altered BCSFB permeability in the male or, on the contrary, underestimating it in the female. We have recently demonstrated that not only albumin but also serum IgG are more concentrated in males than in females CSF [17]. In the present study, we tried to understand if this increased BCSFB permeability to serum IgG can somehow impact the quantitative and qualitative determination of the intrathecal IgG synthesis. The linear IgG index and Reiber’s hyperbolic formula were analyzed, and both formulas showed similar performance in the two sexes without particular differences in concordance, sensitivity, specificity, and positive and negative predictive values compared to the gold standard of CSF-restricted OCB. This finding was found in both MS and CIS patients and confirmed a previous observation seen in a large cohort of neurological patients [33]. We also underline the fact that both quantitative indexes of intrathecal synthesis were negative in 40–50% of MS patients and over 50–60% of CIS patients against a negative rate of about 10% for CSF-restricted OCB IgG in both MS and CIS. This observation confirms that qualitative methods, although more expensive in terms of time, actual reagents, and need for greater operator intervention and preparation, are more sensitive than quantitative quotients to detect intrathecal IgG production [12,21]. Due to such a large number of false negatives, the use of quantitative indexes alone without a parallel OCB research in CSF and serum should be discouraged.

In our two patient populations, each analyzed as a whole, we confirmed the role of age in increasing BCSFB function. However, when patients were grouped by sex, we found that this aging effect was less evident or even absent in women with MS or CIS, respectively. These data confirm that the use of the same thresholds as a function of age for the QAlb values in the two sexes can create criticalities and evaluation disparities [16,18,19].

Age was also correlated to disease duration and severity in overall MS population as well as in male and female subgroups, confirming an effect of aging on disease progression [34]. Despite EDSS being similar in the two sexes at the study time, it is positively correlated to disease duration only in MS patients analyzed as a whole and in the female subgroup but not in the male one. This result could, on the one hand, be due to a lower effect of the duration of the disease on the progression of disability in males, or it could simply be a statistical bias due to the different dimensions of the two subgroups: males and females.

Finally, our results seem to indicate that the IIS, detected through the gold standard of CSF-restricted OCB, was not influenced by either sex or age. Although a lower percentage of positivity emerges in females over 40 years of age (76%) compared to the average of the other subgroups (all close to 90% in MS and greater than 90% in CIS), this reduced positivity in women does not, however, reach statistical significance. Previous studies associated the absence of CSF-restricted OCB with increasing age, without a gender specificity [33,35], or with the male sex irrespective of age [36]. These discrepancies might be due to differences in methodologies (e.g., immunoblotting vs. silver staining), populations (e.g., CIS and MS vs. patients affected by non-inflammatory neurological diseases), and sample size. Future studies based on a larger population of both MS and CIS patients are auspicious to clarify this aspect.

To its observational and retrospective nature, this study has some limitations. First of all, as already pointed out, an analysis of the MRI data is lacking, Furthermore, data on the clinical evolution of the various patients were not collected, and it was therefore not possible to evaluate the long-term effects of the sex-related differences that emerged.

## 5. Conclusions

Taken together, our data confirm the need to review the thresholds used for the assessment of QAlb according to the patient’s sex to avoid a potential underestimation of the BCSFB function in women and/or an overestimation in men. Furthermore, our data show a different role played by age on BCSFB function in the two sexes, with a greater impact on the male sex. Finally, the presence of this sexual dimorphism at the level of CSF protein content does not seem to influence the determination of the intrathecal synthesis of IgG carried out with both quantitative and qualitative methods.

In an era of cost-effective personalized medicine, future studies on the role of sex-related differences in CSF protein content are recommended. In addition, application of a sex-specific approach is advisable for all studies looking for soluble biomarkers, especially those biomarkers that are related to BCSFB function.

## Figures and Tables

**Figure 1 jpm-12-00977-f001:**
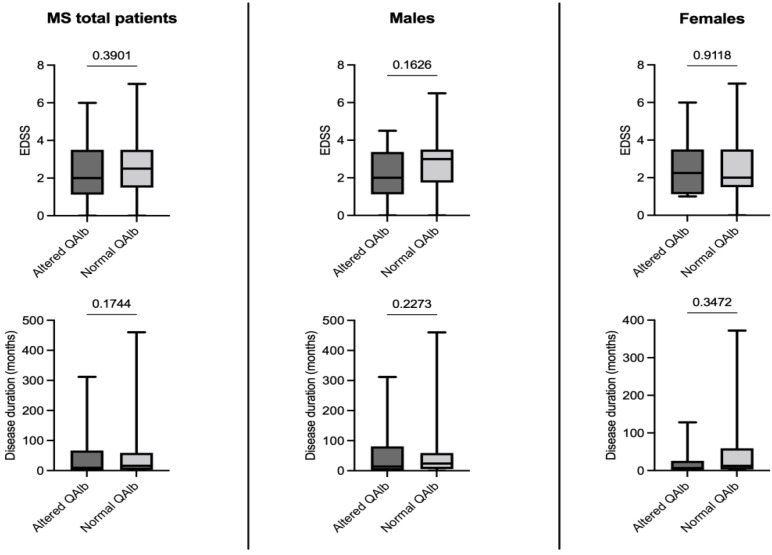
Effect of the blood–cerebrospinal fluid (CSF) barrier permeability evaluated through the albumin quotient (QAlb) on disease severity, expressed with the expanded disability status scale (EDSS), and disease duration in months. Only multiple sclerosis (MS) patients were included (cohort 1 and cohort 2 together). Patients were analyzed as a whole and grouped by sex. Mann–Whitney U test was used for all the comparisons. The boundaries of the boxes represent the 25th–75th quartile. The line within the box indicates the median. The vertical lines above and below the box correspond to the highest and lowest values.

**Figure 2 jpm-12-00977-f002:**
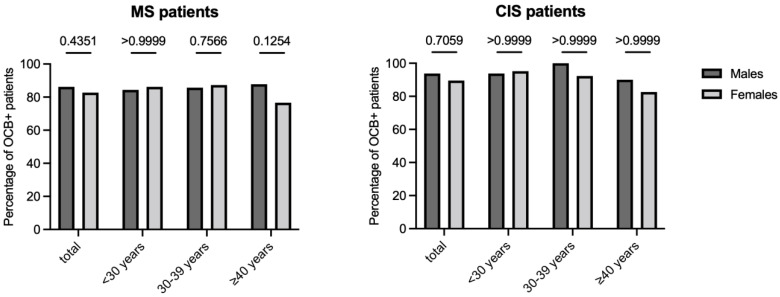
Impact of sex and age on the qualitative determination of an intrathecal IgG synthesis (IIS). The “gold standard” of the cerebrospinal fluid (CSF)-restricted IgG oligoclonal bands (OCB) was used to assess the presence of an IIS. Multiple sclerosis (MS) subjects (cohort 1 and cohort 2 together) and patients with the clinically isolated syndrome (CIS) (cohort 3) were both analyzed as a whole and grouped by age. Fisher’s exact test was used for all comparisons.

**Table 1 jpm-12-00977-t001:** Demographic and clinical characteristics of the study population. Cohort 1 and cohort 2 included patients with multiple sclerosis (MS), while cohort 3 included patients with the clinically isolated syndrome (CIS). Data were referred to the time of lumbar puncture.

	Males	Females	*p*
Cohort 1, MS: Ferrara (Italy): *n*	77	189	
Age, years: median (IQR)	37 (29.0–45.0) ^a^	36 (29.4–44.5) ^b^	0.9270
Clinical activity: *n* (%)	55 (71.4)	143 (75.7)	
Disease duration, months: median (IQR)	27 (3.5–79.5) ^c^	12.5 (3.0–59.75) ^d^	0.1419
EDSS: median (IQR)	3.0 (1.5–3.5) ^e^	2.5 (1.5–3.5) ^f^	0.5651
Cohort 2, MS: San Giovanni Rotondo (Italy): *n*	32	42	
Age, years: median (IQR)	39 (26–46) ^a^	38.5 (28–47) ^b^	0.6080
Clinical activity: *n* (%)	27 (84.4)	35 (83.3)	>0.9999
Disease duration, months: median (IQR)	12 (1–46) ^c^	10 (1–46) ^d^	0.9719
EDSS: median (IQR)	2.0 (1.0–3.0) ^e^	2.0 (1.5–3.0) ^f^	0.8315
Cohort 3, CIS: Ulm (Germany): *n*	32	57	
Age, years: median (IQR)	30 (24–42)	36 (26–46)	0.1660
Disease duration, months: median (IQR)	0 (0–0)	0 (0–6)	>0.9999

Continuous variables were compared with Mann–Whitney U test and categorical variables were compared using Fisher’s exact test. There were no differences between cohort 1 and 2 for age (Mann–Whitney: ^a^
*p* = 0.7292 and ^b^
*p* = 0.8601), disease duration (Mann–Whitney: ^c^
*p* = 0.1185 and ^d^
*p* = 0.3974), and EDSS (Mann–Whitney: ^e^
*p* = 0.1531 and ^f^
*p* = 0.2563). EDSS, expanded disability status scale; IQR, interquartile range.

**Table 2 jpm-12-00977-t002:** Laboratory characteristics of the study population. Cohort 1 and cohort 2 included patients with multiple sclerosis (MS) and were analyzed as a whole, while cohort 3 included patients with the clinically isolated syndrome (CIS).

	Males	Females	*p*
MS patients (Cohorts 1 and 2): *n*	109	231	
QAlb (×1000) value: median (IQR)	5.6 (4.37–8.39) ^a^	4.34 (3.38–5.49) ^b^	<0.0001
QAlb (×1000) max value:	16.3	16.5	
Altered QAlb value: *n* (%)	36 (33.0)	24 (10.4)	<0.0001
Elevated IgG index: *n* (%)	58 (53.2)	140 (60.6)	0.2387
IgG intrathecal fraction > 0: *n* (%)	53 (48.6)	133 (57.6)	0.1304
CSF-restricted IgG OCB: *n* (%)	94 (86.2) ^c^	191 (82.7) ^d^	0.4351
CIS patients (Cohort 3): *n*	32	57	
QAlb (×1000) value: median (IQR)	5.5 (4.7–6.8) ^a^	5.2 (4.05–6.7) ^b^	0.1810
QAlb (×1000) max value:	12.7	11.1	
Altered QAlb value: *n* (%)	9 (28.1)	9 (15.8)	0.1800
Elevated IgG index: *n* (%)	13 (40.6)	26 (45.6)	0.6642
IgG intrathecal fraction > 0: *n* (%)	11 (34.4)	25 (43.9)	0.5002
CSF-restricted IgG OCB: *n* (%)	30 (93.8) ^c^	51 (89.5) ^d^	0.7059

Continuous variables were compared with Mann–Whitney U test, and categorical variables were compared using Fisher’s exact test. No differences were found for QAlb values between ^a^ male groups (Fisher’s exact test: *p* = 0.8731), while ^b^ females affected by CIS had higher QAlb values than females with MS (Fisher’s exact test: *p* = 0.0015). There were no differences between MS and CIS patients for the presence of an intrathecal IgG synthesis (Fisher’s exact test: ^c^ males, *p* = 0.3671; ^d^ females, *p* = 0.2352). CSF, cerebrospinal fluid; IQR, interquartile range; OCB, oligoclonal bands; QAlb, albumin quotient.

**Table 3 jpm-12-00977-t003:** Effect of age on disease duration and severity. Only multiple sclerosis (MS) patients were included (cohort 1 and cohort 2 together). Patients were analyzed as a whole and grouped by sex.

	Age vs. Disease Duration	Age vs. EDSS	EDSS vs. Disease Duration
MS patients (*n* = 340)			
Spearman *r*	0.3502	0.2986	0.2837
95% CI	0.2502–0.4429	0.1955–0.3952	0.1797–0.3815
*p* value	<0.0001	<0.0001	<0.0001
MS males (*n* = 109)			
Spearman *r*	0.4219	0.1997	0.05800
95% CI	0.2487–0.5690	0.006416–0.3786	−0.1371–0.2487
*p*-value	<0.0001	0.0374	0.5491
MS females (*n* = 231)			
Spearman *r*	0.3137	0.3395	0.3798
95% CI	0.1884–0.4289	0.2164–0.4520	0.2598–0.4882
*p*-value	<0.0001	<0.0001	<0.0001

Spearman test was used for all correlations. CI, confidence interval; EDSS, expanded disability status scale.

**Table 4 jpm-12-00977-t004:** Effect of age on quantitative cerebrospinal fluid (CSF) indexes. Albumin quotient (QAlb) was used to quantify blood–CSF barrier permeability. IgG index and Reiber’s intrathecal fraction (IF) formulas were used to assess intrathecal IgG synthesis. Multiple sclerosis (MS) patients were from cohort 1 and cohort 2, while patients with the clinically isolated syndrome (CIS) were from cohort 3. Both groups were analyzed as a whole and divided according to sex.

	Age vs. QAlb (×1000)	Age vs. IgG Index	Age vs. IgG IF (%)
MS patients (*n* = 340)			
Spearman *r*	0.1807	−0.01312	−0.01483
95% CI	0.07270–0.2846	−0.1224–0.09650	−0.1241–0.09481
*p*-value	0.0008	0.8095	0.7853
MS males (*n* = 109)			
Spearman *r*	0.2711 ^a^	0.02820	−0.001166
95% CI	0.08190–0.4415	−0.1662–0.2205	−0.1946–0.1924
*p*-value	0.0043	0.7710	0.9904
MS females (*n* = 231)			
Spearman *r*	0.1540 ^a^	−0.03586	−0.003181
95% CI	0.02161–0.2811	−0.1679–0.09745	−0.1360–0.1297
*p*-value	0.0192	0.5877	0.9616
CIS patients (*n* = 89)			
Spearman *r*	0.2368	0.1775	0.05126
95% CI	0.02380–0.4293	−0.03816–0.3774	−0.1648–0.2626
*p*-value	0.0255	0.0961	0.6333
CIS males (*n* = 32)			
Spearman *r*	0.4561	0.3567	0.1519
95% CI	0.1171–0.6999	−0.001664–0.6338	−0.2180–0.4837
*p*-value	0.0087	0.0451	0.4065
CIS females (*n* = 57)			
Spearman *r*	0.1834	0.1027	−0.01537
95% CI	−0.08886–0.4302	−0.1698–0.3607	−0.2821–0.2536
*p*-value	0.1720	0.4470	0.9096

Spearman test was used for all correlations. ^a^ Differences between slopes were not significant (F = 1.849, *p* = 0.1748), while differences between the elevations were extremely significant (F = 41.71, *p* < 0.0001). CI, confidence interval.

**Table 5 jpm-12-00977-t005:** Impact of sex on the agreement between quantitative and qualitative methods for the determination of an intrathecal IgG synthesis. The “gold standard” of the cerebrospinal fluid (CSF)-restricted IgG oligoclonal bands was used to calculate: (i) concordance through the Cohen’s kappa, (ii) sensitivity, (iii) specificity, and (iv) positive and (v) negative predictive value. Multiple sclerosis (MS) subjects (cohort 1 and cohort 2 together) and patients with the clinically isolated syndrome (CIS) (cohort 3) were both analyzed as a whole and grouped by sex.

	IgG Index	Reiber’s Formula
	Values (95% CI)	Values (95% CI)
MS patients (*n* = 340)		
Kappa	0.371 (0.284–0.458)	0.331 (0.249–0.414)
Sensitivity	0.6807 (0.6245–0.7321)	0.6386 (0.5813–0.6922)
Specificity	0.9273 (0.8274–0.9714)	0.9273 (0.8274–0.9714)
Positive Predictive Value	0.9798 (0.9492–0.9921)	0.9785 (0.9460–0.9916)
Negative Predictive Value	0.3592 (0.2849–0.4408)	0.3312 (0.2617–0.4088)
MS males (*n* = 109)		
Kappa	0.307 (0.174–0.441)	0.234 (0.110–0.359)
Sensitivity	0.617 (0.5160–0.7089)	0.5532 (0.4526–0.6496)
Specificity	1.000 (0.7961–1.000)	0.9333 (0.7018–0.9966)
Positive Predictive Value	1.000 (0.9379–1.000)	0.9811 (0.9006–0.9990)
Negative Predictive Value	0.2941 (0.1871–0.4300)	0.2500 (0.1552–0.3769)
MS females (*n* = 231)		
Kappa	0.407 (0.296–0.518)	0.385 (0.279–0.518)
Sensitivity	0.712 (0.6441–0.7716)	0.6806 (0.6115–0.7426)
Specificity	0.900 (0.7695–0.9604)	0.925 (0.8014–0.9742)
Positive Predictive Value	0.9714 (0.9288–0.9888)	0.9774 (0.9358–0.9939)
Negative Predictive Value	0.3956 (0.3013–0.4983)	0.3776 (0.2879–0.4764)
CIS patients (*n* = 89)		
Kappa	0.143 (0.046–0.240)	0.126 (0.039–0.213)
Sensitivity	0.4815 (0.3760–0.5886)	0.4444 (0.3412–0.5527)
Specificity	1.000 (0.6756–1.000)	1.000 (0.6756–1.000)
Positive Predictive Value	1.000 (0.9103–1.000)	1.000 (0.9036–1.000)
Negative Predictive Value	0.16 (0.08337–0.2851)	0.1509 (0.07852–0.2705)
CIS males (*n* = 32)		
Kappa	0.087 (−0.034–0.209)	0.067 (−0.029–0.164)
Sensitivity	0.4333 (0.2738–0.6080)	0.3667 (0.2187–0.5449)
Specificity	1.000 (0.1777–1.000)	1.000 (0.1777–1.000)
Positive Predictive Value	1.000 (0.7719–1.000)	1.000 (0.7412–1.000)
Negative Predictive Value	0.1053 (0.01870–0.3139)	0.09524 (0.01692–0.2891)
CIS females (*n* = 57)		
Kappa	0.180 (0.042–0.317)	0.168 (0.037–0.299)
Sensitivity	0.5098 (0.3768–0.6414)	0.4902 (0.3586–0.6232)
Specificity	1.000 (0.6097–1.000)	1.000 (0.6097–1.000)
Positive Predictive Value	1.000 (0.8713–1.000)	1.000 (0.8668–1.000)
Negative Predictive Value	0.1935 (0.09187–0.3628)	0.1875 (0.08890–0.3531)

No differences were found between male and female MS patients for sensitivity and specificity of both the IgG index (Fisher’s exact test: *p* = 0.2597 and *p* = 0.5655, respectively) and Reiber’s formula (Fisher’s exact test: *p* = 0.0656 and *p* > 0.9999, respectively). CI, confidence interval.

## Data Availability

The datasets used and analyzed during the current study are available from the corresponding author on reasonable request.

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
