# Peer review of "The Sexual Dimorphism in Cerebrospinal Fluid Protein Content Does Not Affect Intrathecal IgG Synthesis in Multiple Sclerosis"

_jpm, 2022, doi:10.3390/jpm12060977_

Round 1
Reviewer 1 Report
Since cerebrospinal fluid (CSF) analysis with the demonstration of an intrathecal IgG synthesis is the laboratory support for the MS diagnosis and a sexual dimorphism was described recently in CSF protein composition, the authors retrospectively 340 MS patients and 89 people with CIS to assess the impact of sex and age. Doing so, the albumin quotient (QAlb) was shown to be higher in males (5.6 Vs. 4.34) and to correlate to age with a constant gap between sexes. Furthermore, age was shown to positively correlate to disease duration and severity in MS. However, most importantly, this sexual dimorphism seems to not affect the determination of the intrathecal IgG synthesis evaluated by quantitative and qualitative methods.
Major issues:
Lines 374-385 and 400-408: Pannewitz-Makaj et al. 2020 (https://pubmed.ncbi.nlm.nih.gov/33379245/) should be discussed in these paragraphs.
If the data are available, the influence of renal function impairment should be investigated (similar to Konen et al. 2021; https://pubmed.ncbi.nlm.nih.gov/34840504/ ).
Minor issues:
Lines 44-45: “Almost 85% of people with MS have a relapsing-remitting course characterized by clinical relapses alternating with phases of well-being”. Reference required.
Lines 120-122: “For MS patients, relapse was defined as the onset of new or recurrent signs or symptoms, or the worsening of already present neurological abnormalities persisting for at least 24 h in the absence of fever and preceded by at least 1 month of stable or improved neurological state.” Reference required.
Lines 333-336: “In the case of CSF analysis for the classification of the neurological patient, a sex-specific approach to laboratory data has shown in recent years a greater permeability of BCSFB in males compared to females with a consequent increase of protein transferred from the blood to CSF.” Reference required.
The whole manuscript text should be reviewed for minor spelling and punctuation issues (e.g. “males than in females MS patients” (line 207); “Age were also positively…” (line 236))

Author Response
Point-by-point reply to Reviewer: 1
Major issues:
Lines 374-385 and 400-408: Pannewitz-Makaj et al. 2020 (https://pubmed.ncbi.nlm.nih.gov/33379245/) should be discussed in these paragraphs.
We thank the Reviewer for this comment. The manuscript by Pannewitz-Makaj was added to References list, and it was discussed in the text as suggested by the Reviewer.
If the data are available, the influence of renal function impairment should be investigated (similar to Konen et al. 2021; https://pubmed.ncbi.nlm.nih.gov/34840504/ ).
We thank the Reviewer for this point. Unfortunately, data on renal function were not available for all of our patients. However, since our article is not focused on immunoglobulin-free light chains, as opposed to the one mentioned by the Reviewer, we think that this does not constitute a limitation of our study and therefore this aspect has not been taken into further consideration.
Minor issues:
Lines 44-45: “Almost 85% of people with MS have a relapsing-remitting course characterized by clinical relapses alternating with phases of well-being”. Reference required.
We thank the Reviewer for this point. A reference was added as suggested.
Lines 120-122: “For MS patients, relapse was defined as the onset of new or recurrent signs or symptoms, or the worsening of already present neurological abnormalities persisting for at least 24 h in the absence of fever and preceded by at least 1 month of stable or improved neurological state.” Reference required.
We thank the Reviewer. A reference was added as suggested.
Lines 333-336: “In the case of CSF analysis for the classification of the neurological patient, a sex-specific approach to laboratory data has shown in recent years a greater permeability of BCSFB in males compared to females with a consequent increase of protein transferred from the blood to CSF.” Reference required.
We thank the Reviewer also for this suggestion. A reference was added.
The whole manuscript text should be reviewed for minor spelling and punctuation issues (e.g. “males than in females MS patients” (line 207); “Age were also positively...” (line 236)).
We thank the Reviewer for him/her suggestion. The English language was reviewed by a native English teacher from our university, and she was acknowledged in the "Acknowledgment" paragraph.
Reviewer 2 Report
I think the manuscript is well presented and just minor English editing is needed.
Some of the tables could be changed into figures for better illustration and a summary figure would be beneficial to readers.
Author Response
Point-by-point reply to Reviewer: 2
I think the manuscript is well presented and just minor English editing is needed.
We thank the Reviewer for the positive comment and for him/her suggestion. The English language was reviewed by a native English teacher from our university, and she was acknowledged in the Acknowledgment paragraph.
Some of the tables could be changed into figures for better illustration and a summary figure would be beneficial to readers.
We thank the reviewer for this point. Two figures have been added in place of two tables. In particular, Figure 1 has replaced Table 3, and Figure 2 has replaced Table 6.